# Diagnostic and Therapeutic Pathway of Advanced Ovarian Cancer with Peritoneal Metastases

**DOI:** 10.3390/cancers15020407

**Published:** 2023-01-07

**Authors:** Valentina Ghirardi, Anna Fagotti, Luca Ansaloni, Mario Valle, Franco Roviello, Lorena Sorrentino, Fabio Accarpio, Gianluca Baiocchi, Lorenzo Piccini, Michele De Simone, Federico Coccolini, Mario Visaloco, Stefano Bacchetti, Giovanni Scambia, Daniele Marrelli

**Affiliations:** 1UOC Ginecologia Oncologica, Dipartimento per la Salute della Donna e del Bambino e della Salute Pubblica, Fondazione Policlinico Universitario a Gemelli (IRCCS), Catholic University of Sacred Heart, 00167 Rome, Italy; 2Unit of General Surgery, San Matteo Hospital, 27100 Pavia, Italy; 3Peritoneal Tumours Unit, IRCCS Regina Elena National Cancer Institute, 00144 Rome, Italy; 4Unit of General Surgery and Surgical Oncology, Department of Medicine, Surgery, and Neurosciences, University of Siena, 53100 Siena, Italy; 5SC Chirurgia Generale d’Urgenza ed Oncologica, AOU Policlinico di Modena, 41124 Modena, Italy; 6CRS and HIPEC Unit, Pietro Valdoni, Umberto I Policlinico di Roma, 00161 Roma, Italy; 7Department of Clinical and Experimental Sciences, University of Brescia, ASST Spedali Civili, 25123 Brescia, Italy; 8General and Peritoneal Surgery, Department of Surgery, Pisa University Hospital, 56100 Pisa, Italy; 9Candiolo Cancer Institute, FPO-IRCCS, Candiolo, 10060 Torino, Italy; 10General, Emergency and Trauma Surgery, Pisa University Hospital, 56100 Pisa, Italy; 11U.O.C Pronto Soccorso Generale. Con O.B.I., Azienda Ospedaliera Universitaria “G. Martino”, 98124 Messina, Italy; 12AOUD Center Advanced Surgical Oncology, DAME University of Udine, 33100 Udine, Italy

**Keywords:** ovarian cancer, debulking surgery, treatment pathway, diagnosis, neo-adjuvant chemotherapy

## Abstract

**Simple Summary:**

As ovarian cancer represents the most lethal gynecological malignancy, the diagnostic process represents a crucial step in order to select the appropriate treatment strategy. Indeed, the association of tumor marker levels with radiological imaging and an evaluation of tumor load with diagnostic laparoscopy are essential to assess whether the patients are best treated by upfront surgery or neo-adjuvant chemotherapy followed by interval debulking surgery. With this review, we aim to provide most relevant available evidence on the diagnostic and treatment pathway of advanced ovarian cancer.

**Abstract:**

Over two thirds of ovarian cancer patients present with advanced stage disease at the time of diagnosis. In this scenario, standard treatment includes a combination of cytoreductive surgery and carboplatinum–paclitaxel-based chemotherapy. Despite the survival advantage of patients treated with upfront cytoreductive surgery compared to women undergoing neo-adjuvant chemotherapy (NACT) and interval debulking surgery (IDS) due to high tumor load or poor performance status has been demonstrated by multiple studies, this topic is still a matter of debate. As a consequence, selecting the adequate treatment through an appropriate diagnostic pathway represents a crucial step. Aiming to assess the likelihood of leaving no residual disease at the end of surgery, the role of the CT scan as a predictor of cytoreductive outcomes has shown controversial results. Similarly, CA 125 level as an expression of tumor load demonstrated limited applicability. On the contrary, laparoscopic assessment of disease distribution through a validated scoring system was able to identify, with the highest specificity, patients undergoing suboptimal cytoreduction and therefore best suitable for NACT-IDS. Against this background, with this article, we aim to provide a comprehensive review of available evidence on the diagnostic and treatment pathways of advanced ovarian cancer.

## 1. Introduction

Approximately 75% of ovarian cancer patients present with an advanced stage disease at the time of diagnosis. In this scenario, standard treatment includes a combination of cytoreductive surgery and carboplatinum–paclitaxel-based chemotherapy [1]. Indeed, those women are usually treated with either primary debulking surgery (PDS) followed by adjuvant platinum-based chemotherapy or with neo-adjuvant chemotherapy (NACT) prior to interval debulking surgery (IDS) and further postoperative chemotherapy [2]. Bearing in mind that no gross residual disease (NGR) remains the main goal of both upfront and interval debulking surgery [2,3], a survival advantage of patients treated with primary cytoreductive surgery compared to women undergoing NACT-IDS has been demonstrated by multiple studies [4,5,6]. Although not without flaws and biases, the non-inferiority in terms of survival of the latter treatment modality compared to PDS has been prospectively demonstrated [7,8]. Due to that, some authors support its use to minimize the patient’s morbidity in relation to the reduced surgical effort, especially in case of high tumor load [9]. In order to select the appropriate treatment pathway according to the patient’s and tumor’s characteristics, the diagnostic process represents a crucial step. In advanced ovarian cancer it consists in a combined evaluation of imaging techniques, serum level of tumor markers, and the patient’s fitness for surgery, together with intraoperative assessment of disease spread. Against this background, with this article, we aim to provide a comprehensive review of available evidence on the diagnostic and treatment pathways of advanced ovarian cancer.

## 2. Sources and Methodology

The review of the literature included articles published from inception until May 2022. The search was performed in the Pubmed and Embase databases and included the combination of the following medical subject headings (MeSH): ‘ovarian cancer’ & ‘diagnosis’, ‘laparoscopy’, ‘surgery’, ‘patient’s selection’, ‘chemotherapy’. Review articles, books and monographs were also consulted. All pertinent manuscripts were included, prioritizing randomized controlled trials (RCT), meta-analyses, observational studies, consensus statements, and systematic reviews. Publications within the past decade were prioritized, as well as articles considered as landmarks in the treatment paradigm of ovarian cancer. Only papers published in English were reviewed.

## 3. Diagnostic Pathway

### 3.1. Radiologic Assessment and Role of Tumor Markers

The main objective of preoperative assessment in advanced ovarian cancer patients is to predict cytoreductive outcomes, specifically to determine the likelihood of leaving NGR at the end of debulking surgery [10]. In this context, the radiologic assessment of disease distribution and the evaluation of tumor markers levels represent one of the first steps in the diagnostic process. In ovarian cancer literature, the role and the efficacy of the computed tomography (CT) scan as a surgical outcome predictor have been widely investigated, with controversial results [11,12,13]. A radiologic predictive model of the optimal cytoreductive rate based on CT scan findings in advanced ovarian cancer was developed by Bristow et al. [11]. In their paper, CT scan findings of 41 patients were retrospectively analyzed and several CT scan features (peritoneal thickening, peritoneal implants, bowel mesentery involvement, suprarenal paraaortic lymph nodes, omental extension, and pelvic sidewall involvement and/or hydroureter) appeared to be associated with the surgical outcome. In their predictive model, the ability to predict surgical outcome was statistically significant (*p* < 0.001), with an identification of patients undergoing optimal and suboptimal cytoreduction with a sensitivity of 85% and 100%, respectively. On the other hand, results coming from the above-mentioned trial [11] could not be confirmed in their cross validation with two other models of CT scan prediction of cytoreductive outcomes [12], with a dropping of the accuracy rate of the model by Bristow et al. [11] from 93% to 74%. According to the results by Axtell et al. [12], preoperative CT scan determinants should be used with caution when planning a treatment strategy. Alongside that, another predictive model was prospectively created, analyzing data from over 600 advanced ovarian cancer patients [13]. At multivariate analysis, three clinical and six radiologic criteria were significantly associated with suboptimal debulking and assigned to a value score, which showed a predictive accuracy of 0.758. Of note, among the identified clinical features, a CA 125 level > 500 U/mL was independently associated with suboptimal debulking surgery (*p* < 0.001) [13]. Indeed, the above-mentioned cut-off for the CA 125 level has been used by most of the researchers as an expression of tumor burden and consequently related to tumor resectability. However, studies on the association of the CA 125 level and surgical outcomes have shown controversial results [13,14,15] and overall demonstrated a limited applicability in preoperative treatment planning. Indeed, results from a meta-analysis on this topic [16], demonstrated CA 125 to be a strong risk factor for suboptimal cytoreduction but with low ability to accurately predict optimal cytoreduction. In conclusion, in preoperative treatment planning, the utility of both CT scan findings and CA 125 level is limited. However, the power of radiologic assessment can be increased by its association with diagnostic laparoscopy, which showed a decreasing of nearly 60% of unnecessary laparotomies on 350 prospectively analyzed patients [17].

### 3.2. Selection of the Appropriate Surgical Candidate

Despite there being now many tools able to assist the gynecologic oncologist to assess disease resectability [18,19], not as many data are yet available to help in selecting the adequate surgical candidate for cytoreductive surgery. Indeed, ovarian cancer patients are a population that continues to age, with generally impaired nutritional status and physical conditions which can both affect fitness for surgery. Even if chronologic age alone cannot be considered a risk factor for poor recovery after surgery [1], it contributes to determine a patient’s frailty, which is to be considered a syndrome able to affect capabilities to maintain homeostasis after a stressor [20] and appears to be associated with adverse survival outcomes [21]. Generally, a patient’s frailty is defined by either the physical phenotypic model as stated by Fried et al. [22], or the deficit accumulation model, called Frailty Index (FI), which measures the total of deficits by assessing disease status, symptoms and signs together with disabilities in daily life [23]. Parallel to a limited number of retrospective studies demonstrating the negative impact of frailty on surgical and survival outcomes [24,25], prospective data coming from a recent evaluation of the association of FI [23] and postoperative complications together with overall survival in 144 ovarian cancer patients was conducted by Inci et al. [26]. They found that for patients with FI > 0.26 [23] (33% of cases), the risk of developing severe post-operative complications [27] was five times higher than the rest of the population (OR 4.74, 95% CI: 1.96–11.53, *p* = 0.001). In addition, as well as the presence of residual tumor > 1 cm and low albumin level (<3.5 g/dL), FI < 0.15 [24] was associated with poor overall survival (OS). In the current literature, it is becoming more evident that the assessment of a patient’s frailty is already a determinant in the decision-making process of treatment strategy, also influencing both patient’s participation in clinical trials [1] and the administration of additional treatments, such as hyperthermic intraperitoneal chemotherapy [28]. Alongside the identification of frail patients, an attempt to provide a reproducible and standardized treatment selection tool has been provided by Narasimhulu et al. [29], who tested if their evidenced-based algorithm was able to reduce morbidity and mortality related to debulking procedure. Indeed, they identified a patient as high risk for adverse events after surgery if at least one of the following high-risk criteria were present: (i) albumin < 3.5 g/dL, (ii) age ≥ 80 years, or (iii) age 75–79 and at least one of the following: ECOG (Eastern Cooperative Oncology Group) performance status >1, stage IV disease or complex surgery likely to be required. In the analyzed population of 334 advanced ovarian cancer patients undergoing either PDS or NACT/IDS according to the above-mentioned triage strategy, 70% of included patients were offered upfront surgery with morbidity, mortality and complete resection rate comparable to the IDS group. However, OS was shown to be superior in the group selected for PDS regardless residual disease status, with three-year estimates OS of 72.5% vs. 50.0% (95% for the IDS group (*p* = 0.007) in case of NGR. To conclude, in a scenario where the treatment paradigm is progressively moving from the dichotomy PDS-IDS to the personalization of treatment, an appropriate assessment of patient’s fitness and overall performance status is imperative to adequate plan treatment strategy, in order to put in place adequate pre-habilitation programs. 

### 3.3. The Role of Laparoscopy in Treatment Selection

The use of laparoscopy as a valid tool to assess disease burden and predict disease resectability is now well recognized and accepted by both ESMO-ESGO [30] and National Comprehensive Cancer Network (NCCN) guidelines [31]. With the final aim to create an objective and standardized disease assessment model, the Predictive Index Value (PIV) was designed in 2006 by Fagotti et al. [18]. In their study, 64 advanced ovarian cancer patients were submitted to both laparoscopy and longitudinal laparotomy to define the chances of optimal debulking. Seven laparoscopic parameters were identified and associated to a numerical variable in relation to the strength of statistical association. In the final model, a predictive index score > or =8 identified patients undergoing suboptimal surgery with a specificity of 100%. The positive predictive value (PPV) was 100%, and the negative predictive value (NPV) was 70%. Along with the further validation in 2015, after the introduction of upper abdominal surgery [32] which confirmed an overall accuracy of the PIV [18] ≥60% in all the 6 parameters and a PPV = 100% for complete disease unresectability in case of PIV ≥ 10 [18], the concordance of the scoring algorithm with pattern of disease distribution identified at PDS was retrospectively assessed in 226 patients submitted to both diagnostic laparoscopy and open exploration of the abdominal cavity [33]. A 96% of overall concordance between the two assessments was identified and laparoscopic assessment of the abdominal cavity was considered able to predict NGR in advanced ovarian cancer [33]. The same laparoscopic selection method can be applied to the IDS setting. Indeed, a modified PIV score [34] was prospectively created and published in 2010 in order to help identifying patients suitable for complete cytoreduction after NACT. In the final calculation, four out of the six variables of the PIV score [18] were included and used to create the final model with the same statistical process. In this setting, a PIV > 4 corresponds to a probability of optimally resecting the disease at IDS equal to 0, therefore surgery should be abandoned. As mentioned before, the use of the CT scan to assess tumor burden may benefit from the integration with laparoscopy in order to increase its accuracy. On this topic, the association of CT and laparoscopic data to determine peritoneal cancer index (PCI) together with lesion size score was evaluated in the R3 and R4 model scores [35]. The efficacy of laparoscopy in assessing disease distribution and to predict NGR was subsequently confirmed in a population of 103 ovarian cancer patients where the three above-mentioned scores (Fagotti score [18], R3 and R4 models [35]) were retrospectively analyzed [19]. Future perspectives on diagnostic and treatment pathway algorithm for advanced ovarian cancer is depicted in Figure 1.

## 4. Treatment Pathway

When preoperative assessment of patient’s fitness and disease burden is completed, treatment choice may consist in either upfront PDS or NACT-IDS. In general, when patient’s conditions and tumor distribution allow it, primary cytoreductive surgery followed by platinum-based chemotherapy remains the standard treatment for patients with advanced stage ovarian cancer [36], as it has been demonstrated to prolong patient’s survival compared to the combination NACT-IDS [4,5,6]. Due to that, NACT may be considered for patients deemed unlikely to be completely cytoreduced to NGR or for patients who are poor surgical candidates [36], for which an ultrasound (US)-guided biopsy can represent a valid alternative to achieve final histology [37]. Of note, when preoperatively available, tumor histotype may play a crucial role in the treatment selection process. Indeed, it has been widely demonstrated that some ovarian cancer histotypes/grades have a low response rate to chemotherapy, therefore upfront surgery may remain an option in selected cases despite the unfavorable tumor burden [38,39,40]. Specifically, in case of low-grade serous ovarian cancer, the survival benefit of NACT appears to be less evident compared to high grade serous cancer. Indeed, in those patients an extensive cytoreductive surgery represents the best option to provide a survival advantage even in case of high tumor load. Due to that, surgery in those cases often implies a higher surgical complexity score and subsequently more extensive surgical procedures, which in some cases may lead to an increased peri-operative morbidity rate [41]. Similar considerations can be made in case of mucinous histology as those patients present an overall poor response to chemotherapy and significantly worse survival with respect to patients with different histology [42]. 

In view of those considerations, the SGO and American Society of Clinical Oncology (ASCO) clinical practice guidelines state that in case of a high likelihood of achieving NGR or residual disease less than 1 cm with acceptable morbidity, PDS should be the preferred treatment choice (evidence quality: intermediate; strength of recommendation: moderate) [27]. Despite this, the better outcome of PDS-treated patients over those undergoing NACT at the time of primary treatment is still a matter of debate. On this topic, few phase III trials have investigated whether NACT-IDS is equally effective and safe as PDS followed by platinum based chemotherapy in this population [4,7,8,9]. In the (EORTC)-55971 trial [8], 670 women with FIGO stage IIIC/IV [43] EOC were randomized to NACT-IDS versus PDS, showing a superimposable median OS between the two groups (29 vs. 30 months; respectively) but with lower surgical-related morbidity for NACT-treated patients. The same non-inferiority results in terms of survival outcomes of NACT-IDS-treated patients were achieved in the CHORUS trial [7], which equally randomized 550 advanced ovarian cancer cases in the same two treatment arms, showing again comparable OS between the two groups (22.8 months for PDS group vs. with 24.5 months for NACT-IDS). Later, a pooled analysis of individual patient data included in EORTC 55971 [8] and CHORUS trial [7], demonstrated improved survival for FIGO stage IV [43] patients treated with NACT-IDS [44] (median OS 24.3 months in NACT-IDS versus 21.2 months in the PCS group *p* = 0.48; and median PFS 10.6 versus 9.7 months *p* = 0.049). Despite their prospective nature, few criticisms were raised to the results of the above-mentioned trials, mainly related to patient’s selection bias and low NGR rate which could have affected survival estimation. In addition to that, results supporting the use of NACT with respect to PDS in case of high tumor load are coming from the recently published SCORPION trial [9]. In this study, 171 advanced ovarian cancer patients with a 8 < PIV < 12 [18] were randomly assigned to receive either PDS or NACT-IDS, demonstrating a comparable survival between the two groups but with different surgical morbidity profile in favor of NACT-IDS arm (Median PFS 15 and 14 months, *p* = 0.73, respectively; median OS 41 and 43 months, respectively, *p* = 0.56). Currently, results from an ongoing phase III trial [45] on this topic are awaited to further assess OS of advanced ovarian cancer patients submitted either to PDS or NACT-IDS, hopefully overcoming the limitations highlighted in the previous trials. It has, however, to be mentioned that whenever surgery is performed, maximal surgical effort to achieve NGR is imperative as it has been demonstrated that the volume of residual disease is one of the most powerful determinants of survival in both upfront and interval cytoreductive setting [46,47,48]. As additional treatment at the time of IDS, together with the removal of all visible disease, cisplatin 100 mg/m^2^ as hyperthermic intraperitoneal chemotherapy (HIPEC) can be delivered to further improve oncologic outcomes. Indeed, results form a prospective randomized study on this topic [49] showed an improving in both PFS and OS in the HIPEC treatment arm vs. standardly treated patients (PFS 14.2 vs. 10.7, *p* = 0.003 respectively; OS 45.7 vs. 33.9, *p* = 0.02 respectively) without an increased procedure-related morbidity [50]. Survival outcomes on the use of HIPEC with the same chemotherapy regimen at the time PDS is currently under investigation and results are expected by 2026 [51]. Alongside that, because targeting a usually intraperitoneal disease with an intraperitoneal treatment seems to represent a promising option in advanced ovarian cancer and in view of the controversies which arose among the experts, mostly in relation to HIPEC morbidity data [52], few trials are currently ongoing, with the aim to further assess the positive results achieved at the time of IDS and to potentially expand the applicability of HIPEC in other treatment settings [53]. Indeed, currently available RCTs failed to demonstrate a survival improvement in HIPEC-treated patients both at the time PDS [54] and disease recurrence [55] with respect to only surgically treated patients.

Overall, a recent meta-analysis on current available evidence on HIPEC associated with cytoreductive surgery in ovarian cancer showed an improvement in both PFS (HR, 0.585; 95% CI, 0.422–0.811) and OS (HR, 0.519; 95% CI, 0.346–0.777) in case of recent exposure to chemotherapy (<6 months) whilst no survival benefit was detected in patients submitted to chemotherapy > 6 months before HIPEC administration (HR, 1.037; 95% CI, 0.684–1.571; HR, 0.932; 95% CI, 0.607–1.430, respectively) [56]. Moreover, as precision medicine is rapidly expanding in ovarian cancer treatment and because performance of tumor genetic assessment is leading to an increasing number of molecular-driven therapies, interesting data are coming on the influence of BRCA/HRD molecular status on HIPEC efficacy in this subset of patients. On this topic, data published by Koole et al. [57]. showed that patients with homologous recombination deficiency (HRD) tumors without pathogenic BRCA1/2 mutation appear to benefit the most from treatment with HIPEC, while the benefit in patients with BRCA1/2 pathogenic mutations and patients without HRD tumors seems less evident. In addition, despite not yet recruiting, the GOG3068 aims to prospectively randomize 230 patients to receive or not receive HIPEC with cisplatin 100 mg/m^2^ at the time of IDS followed by adjuvant chemotherapy and maintenance treatment with Niraparib 300 mg/daily, with PFS analysis as primary endpoint and OS stratified for both RD after surgery and HRD status as secondary outcome.

In terms of adjuvant chemotherapy regimen in primary treatment setting, improved OS in patients treated with paclitaxel other than cyclophosphamide in addition to cisplatin has been shown in the GOG111, a phase III trial [58]. As carboplatin appeared to be as effective as cisplatin but with less side effects [59] and that weekly-regimen does not add any additional benefits compared to three-weekly administration [60], carboplatin and paclitaxel based chemotherapy administered every 21 days has become the standard of treatment in both the neo-adjuvant and adjuvant setting [30]. As maintenance therapy, a modest improvement in PFS only (10.3 months in the control group vs. 14.1 in the bevacizumab group) in patients treated with angiogenesis inhibitors was described in two prospective randomized trial, the ICON7 and GOG0218 [61,62]. Due to that and to its acceptable toxicity profile, the use of Bevacizumab as maintenance therapy in advanced EOC was approved by the US Food and Drug Administration (FDA) in 2018. More impressive results in terms of survival gain are coming from vascular endothelial growth factor (VEGF) inhibition through the use of Poly ADP-ribose polymerase inhibition (PARPi) as maintenance therapy in advanced ovarian cancer patients. Indeed, in the SOLO-1 trial [63] a phase III randomized multicenter trial, the efficacy of Olaparib as maintenance monotherapy compared with placebo was evaluated in 391 patients with newly diagnosed advanced BRCA mutated ovarian cancer following platinum-based chemotherapy. The risk of disease progression or death was reduced by the use of Olaparib by 70% (hazard ratio 0.30, 0.23 to 0.41; *p* < 0.001), and median PFS was not reached after 41 months of follow-up in the Olaparib treated group, compared with 13.8 months for the control group. Alongside and regardless of BRCA mutation, PARPi have shown to be effective in prolonging patients’ survival in case of HRD. Indeed, Velaparib and Niraparib demonstrated in two randomized phase III trials [64,65] a significant PFS gain in comparison to placebo in patients with newly diagnosed advanced epithelial ovarian cancer, further confirming the importance of genetic tumors’ and patients’ assessment to adequately plan maintenance treatment after chemotherapy.

For what concerns the treatment pathway in case of disease recurrence, the role of secondary cytoreductive surgery (SCS) as the best way to achieve a good survival outcome has been demonstrated by multiple studies, both retrospective and prospective [66]. Overall, as in the primary setting, selection of the best surgical candidate represents a crucial aspect, being surgery to NGR, the primary aim also in case of SCS. In this regard, features of patients best suited for surgery in case of disease relapse were performed by Harter et al. [67] in a multicenter retrospective evaluation of 267 patients. In their study, they demonstrated that women undergoing surgery to NGR had significantly better PFS and OS (median OS, 45.2 months vs. 19.7 months for patients with residuals > 10 mm; *p* < 0.0001). Furthermore, the presence of >500 mL of ascites and complete cytoreduction at the time of primary surgery were found to be independent prognostic factors at multivariate analysis. Overall, the combination of these two features, together with the assessment of the patient’s performance status, constitutes the so-called AGO score, which helps to identify patients who will most likely achieve NGR at the time of SCS. The validity of the AGO-score criteria was subsequently prospectively assessed in the DESKTOP II trial [68], which showed a 76% complete resection rate among 216 recurrent ovarian cancer patients who were classified as AGO-score positive, with a perioperative mortality rate < 1%.

More recently, the survival benefit of AGO-score positive platinum-sensitive recurrent ovarian cancer patients undergoing SCS followed by chemotherapy, compared to patients receiving chemotherapy alone, was demonstrated in a prospective randomized trial [69], which showed a median OS of 53.7 months vs. 46.0 months, respectively (*p* = 0.02). In addition, patients with NGR at the end of surgery had the most favorable outcome, with a median OS of 61.9 months. Superimposable results were achieved in another RCT comparing platinum-sensitive relapsed ovarian cancer patients receiving SCS + adjuvant chemotherapy vs. chemotherapy alone [70]. Again, a PFS gain of more than 5 months was detected in patients submitted to SCS + adjuvant chemotherapy (17.4 months vs. 11.9 months, respectively; *p* < 0.0001), further confirming the important role of cytoreductive surgery at the time of disease recurrence as a provider of longer survival.

On the other hand, different results were shown in RCT by Coleman et al. [71], which failed to demonstrate the benefit of the combination of SCS and chemotherapy vs. chemotherapy only among 485 platinum-sensitive recurrent ovarian patients, randomly assigned to one of the two treatment arms. Indeed, their data demonstrated a median OS of 50.6 months and 64.7 months, respectively, with a complete resection rate of 63% and 84% of patients receiving bevacizumab as maintenance treatment, with 11% of patients with a previous bevacizumab exposure. However, the interpretation of the data appears to be limited by the lack of a standardized model to select patients for surgery and by the inclusion of bevacizumab in the treatment design.

Differently from the secondary recurrent setting, the role of cytoreductive surgery in prolonging survival in case of tertiary disease recurrence is still unclear, as evidence is coming from retrospective studies [72,73]. Due to that, data from RCT would be needed to characterize the role of surgery in this subset of patients.

Adjuvant therapy in case of disease recurrence is a combination of platinum-based chemotherapy (recommended regimen in case of platinum-sensitive disease,)and maintenance therapy, which is decided in relation to molecular tumor profiling. Indeed, both BRCA and HRD mutational status are of crucial importance in order to prioritize between competing PARPi and non-PARPi regimens [74].

## 5. Conclusions

In advanced ovarian cancer, the diagnostic pathway includes assessment of whole body imaging, patient’s fitness and intraoperative evaluation of disease spread. Standard surgical treatment consists of PDS followed by adjuvant, three weekly carboplatinum–paclitaxel chemotherapy or the combination of NACT-IDS, in case of unresectable disease or patient’s poor performance status.

Cytoreductive surgery to NGR plays a crucial role to improve patient’s survival also at the time of disease recurrence. As in current times, the ovarian cancer treatment journey is shifting towards personalization of treatment with promising results, studies able to characterize tumor and patient’s biomarkers amenable of targeted and tailored therapies are encouraged.

## Figures and Tables

**Figure 1 cancers-15-00407-f001:**
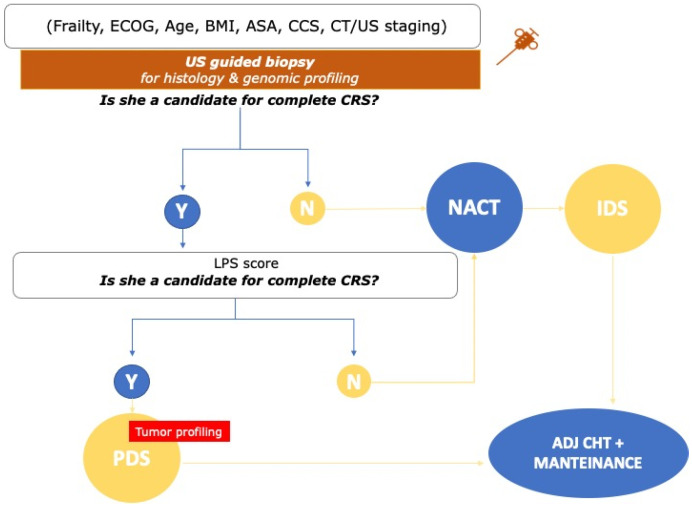
Future perspectives on selection of advanced ovarian cancer patients for primary surgery.

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
