# Peer review of "Diagnostic and Therapeutic Pathway of Advanced Ovarian Cancer with Peritoneal Metastases"

_cancers, 2023, doi:10.3390/cancers15020407_

Round 1
Reviewer 1 Report
The review titled:
Diagnostic and therapeutic pathway of advanced ovarian cancer with peritoneal metastases by Valentina Ghirardi et al. is a comprehensive review. The review article would benefit from a figure and table of treatments and outcomes to emphasize need for individual treatment.
Author Response
Reviewer #1: Diagnostic and therapeutic pathway of advanced ovarian cancer with peritoneal metastases by Valentina Ghirardi et al. is a comprehensive review. The review article would benefit from a figure and table of treatments and outcomes to emphasize need for individual treatment.
Answer: Thank you for your comment. Figure 1 which depicts diagnostic and treatment pathway for advanced ovarian cancer has been implemented , as asked.
Reviewer 2 Report
This is a well written, concise paper that gives the reader an updated overview over the topic of selecting the right treatment for patients with advance ovarian cancer.
However, the discussion and the conclusion do not really add that many novelties to already existing reviews. The authors have a tendency to cite papers within a narrow group of authors (including themselves), and a broader approach might have been of value. By adding a more detailed discussion concerning, for example, the treatment choice in different histologic ovarian cancer types, or BRCA pos/neg patients, the paper would be more forward-looking, in my opinion. Further, surgery in recurrent ovarian cancer was not mentioned.
Author Response
Reviewer #2: This is a well written, concise paper that gives the reader an updated overview over the topic of selecting the right treatment for patients with advance ovarian cancer.
However, the discussion and the conclusion do not really add that many novelties to already existing reviews. The authors have a tendency to cite papers within a narrow group of authors (including themselves), and a broader approach might have been of value. By adding a more detailed discussion concerning, for example, the treatment choice in different histologic ovarian cancer types, or BRCA pos/neg patients, the paper would be more forward-looking, in my opinion. Further, surgery in recurrent ovarian cancer was not mentioned.
Answer: Thank you for your interesting comment and for your important suggestions. The discussion on treatment pathway has been implemented with details on tailoring of treatment in relation to tumor histology (page7, lines 235-243) and more recent evidence on association of HIPEC treatment with tumor genetic profiling, with an overall implementation of manuscript references (page 8, line 285-310).. Also, a section of the discussion focused on recurrent disease has been added, as requested (page 9 from line 338).